# Genome Editing for the Understanding and Treatment of Inherited Cardiomyopathies

**DOI:** 10.3390/ijms21030733

**Published:** 2020-01-22

**Authors:** Quynh Nguyen, Kenji Rowel Q. Lim, Toshifumi Yokota

**Affiliations:** 1Department of Medical Genetics, Faculty of Medicine and Dentistry, University of Alberta, Edmonton, AB T6G2H7, Canada; nguyenth@ualberta.ca (Q.N.); kenjirow@ualberta.ca (K.R.Q.L.); 2The Friends of Garrett Cumming Research & Muscular Dystrophy Canada, HM Toupin Neurological Science Research Chair, Edmonton, AB T6G2H7, Canada

**Keywords:** dilated cardiomyopathy (DCM), hypertrophic cardiomyopathy (HCM), restrictive cardiomyopathy (RCM), arrhythmogenic right ventricular cardiomyopathy (ARVC), left ventricular non-compaction cardiomyopathy (LVNC), Duchenne muscular dystrophy, dystrophin, genome editing, CRISPR/Cas9, Cpf1 (Cas12a)

## Abstract

Cardiomyopathies are diseases of heart muscle, a significant percentage of which are genetic in origin. Cardiomyopathies can be classified as dilated, hypertrophic, restrictive, arrhythmogenic right ventricular or left ventricular non-compaction, although mixed morphologies are possible. A subset of neuromuscular disorders, notably Duchenne and Becker muscular dystrophies, are also characterized by cardiomyopathy aside from skeletal myopathy. The global burden of cardiomyopathies is certainly high, necessitating further research and novel therapies. Genome editing tools, which include zinc finger nucleases (ZFNs), transcription activator-like effector nucleases (TALENs) and clustered regularly interspaced short palindromic repeats (CRISPR) systems have emerged as increasingly important technologies in studying this group of cardiovascular disorders. In this review, we discuss the applications of genome editing in the understanding and treatment of cardiomyopathy. We also describe recent advances in genome editing that may help improve these applications, and some future prospects for genome editing in cardiomyopathy treatment.

## 1. Introduction

Inherited cardiomyopathies are major causes of cardiac-related morbidity and mortality in all age groups [1]. Based on functional and morphological features, cardiomyopathies are generally classified into five groups: dilated cardiomyopathy (DCM), hypertrophic cardiomyopathy (HCM), restrictive cardiomyopathy (RCM), arrhythmogenic right ventricular cardiomyopathy (ARVC), and left ventricular non-compaction cardiomyopathy (LVNC) [2]. Although significant heterogeneity exists within each group, these categories remain clinically useful in terms of predicting major complications or developing approaches for treatment. With molecular genetics, multiple genes responsible for the development of cardiomyopathies have been identified [1]. Interestingly, different mutations in the same gene can lead to different clinical presentations, and hence different cardiomyopathies [2,3,4].

Cardiomyopathy is also a major manifestation in a subset of neuromuscular disorders [5,6]. Cardiomyopathy is most commonly associated with Duchenne muscular dystrophy (DMD) and Becker muscular dystrophy (BMD), both X-linked recessive disorders caused by mutations in the dystrophin (*DMD*) gene [6]. *DMD* codes for dystrophin, a cytoskeletal protein that maintains the structural integrity of muscle fibers during contraction-relaxation cycles. Loss-of-function mutations in the *DMD* gene result in an absence of dystrophin, leading to DMD [7,8,9]. Mutations maintaining the *DMD* reading frame produce truncated but partly functional dystrophin, and often result in a milder form of the disease known as BMD [10,11]. Cardiomyopathy is a leading cause of death in both DMD and BMD patients and is routinely described as being DCM [11,12]. Other neuromuscular disorders associated with cardiomyopathy include the limb girdle muscle dystrophies, myotonic dystrophy, and Friedreich ataxia [5].

Advances in molecular genetics enable genome editing to be incorporated into various fields of biomedical research, including the cardiovascular sciences. Currently, the most commonly used tools are zinc finger nucleases (ZFNs), transcription activator-like effector nucleases (TALENs), and clustered regularly interspaced short palindromic repeats (CRISPR) [13]. Both ZFNs and TALENs are engineered restriction enzymes created by combining a DNA binding domain with a DNA cleavage domain adapted from the FokI restriction enzyme [14,15,16]. The DNA binding domain in ZFNs is adapted from the classic zinc finger transcription factors and consists of three–six zinc finger repeats that recognize 9–18 bp [14]. The DNA binding domain in TALENs is composed of 10–30 repeats of 33–35 amino acids. The amino acid sequence is highly conserved with the exception of two amino acids, allowing TALENs to recognize specific DNA sequences [17,18,19]. These DNA binding domains enable ZFNs and TALENs to have target specificity. Binding of two independent nucleases on opposite DNA strands allows dimerization of the FokI domains and subsequent double-strand DNA cleavage with sticky ends [16,20].

On the other hand, CRISPR is a genome-editing tool adapted from the bacterial adaptive immune system [21]. There are two major components of CRISPR, namely the guide RNA (gRNA) and CRISPR-associated (Cas) protein [21,22,23,24,25]. gRNA is a roughly 100 nucleotide-long RNA sequence that guides the Cas protein to a specific location in the genome. The first 20 nucleotides of the gRNA (protospacer) can be engineered to hybridize to a specific site on DNA, thereby conferring sequence-specificity to the CRISPR/Cas system. Cas protein also needs to bind to a protospacer-adjacent motif (PAM) sequence, which is species-specific. Once the gRNA protospacer hybridizes to a complementary DNA sequence and the Cas protein binds to a PAM site, the Cas protein cleaves both strands of the DNA resulting in a blunt-ended double-stranded DNA break. While this process holds true for CRISPR/Cas9, the most popular version of CRISPR in the literature, other systems such as CRISPR/CRISPR from Prevotella and Francisella 1 (CRISPR/Cpf1) are capable of creating sticky-end double-stranded DNA breaks [26].

Each of the three genome editing tools has been used in various fields to further our knowledge on the pathophysiology of certain diseases as well as facilitate the development of novel therapies [13,27]. As we will see, however, CRISPR appears to be the most widely used of the three technologies and will thus be given more focus in the ensuing discussion. This review aims to describe the genetics of cardiomyopathies and the diverse applications of genome editing in the understanding and treatment of cardiomyopathy. We end with a brief survey of recent advances in genome editing that may help facilitate cardiomyopathy research, as well as some future prospects for genome editing in the field.

## 2. Cardiomyopathies

### 2.1. Dilated Cardiomyopathy

DCM is the common final phenotype of various heart conditions and the most common indication for cardiac transplantation [1,12,28]. DCM is characterized by left ventricle (LV) dilatation and systolic dysfunction. Inherited DCM is accountable for 30%–50% of cases, with the most common mode of inheritance being autosomal dominant [29,30]. Autosomal recessive, X-linked and mitochondrial inheritance patterns have been described as well, despite being less common [12,28]. Disease-causing mutations have been identified in more than 40 genes, which serve functions in a variety of cellular processes and structures. Defects in components of the nuclear envelope proteins (e.g., lamin A and C), contractile apparatus (e.g., myosin heavy chain beta), membrane scaffolding (e.g., sarcoglycan), calcium handling proteins (e.g., phospholamban) or transcriptional and splicing machinery (e.g., ribonucleic acid-binding protein) have all been implicated in the pathogenesis of DCM [1]. Given the molecular complexity of DCM, multiple factors likely contribute to contractile dysfunction and eventually cardiomyocyte death and myocardial fibrosis, all of which are hallmarks of DCM.

### 2.2. Hypertrophic Cardiomyopathy

HCM is an autosomal dominant condition characterized by concentric hypertrophy of the LV, often with prominent septal hypertrophy [1,12]. Multiple genes encoding sarcomeric proteins are involved in the pathogenesis of HCM. Of these, mutations in *MYH7* (encoding the β-myosin heavy chain) and *MYBPC3* (encoding the cardiac myosin binding protein C) are the most common, each accounting for 25%–30% of HCM cases [31]. Mutations identified in sarcomeric genes typically exert a dominant-negative or haploinsufficiency effect resulting in increased myofilament activation, cardiomyocyte hypercontractility and excessive energy use [32,33]. Alterations in Z-disc proteins (e.g., MLP), calcium-handling proteins (e.g., troponin), proteins involved in myocardial energy generation/energy-sensing (e.g., AMP-activated protein kinase) and in various signaling pathways (e.g., the Janus-associated kinase-signal transducers and activators of transcription [JAK-STAT] signaling pathway) have also been found in HCM [1]. These changes often lead to decreased myocyte relaxation and increased myocyte growth with prominent involvement of the interventricular septum.

### 2.3. Arrhythmogenic Right Ventricular Cardiomyopathy

The main feature of ARVC is fibro-fatty infiltration of the myocardium, mainly in the right ventricle (RV), but the LV can also be involved [34]. ARVC is commonly inherited as an autosomal dominant disorder with incomplete penetrance [2,35,36]. Alterations in genes encoding desmosomal proteins have been identified as disease-causing in ARVC. Desmosomal proteins can be categorized into three groups: transmembrane proteins such as desmosomal cadherins, proteins that attach directly to intermediate filaments such as desmoplakin, and linker proteins such as the armadillo family of proteins. Mutations in genes encoding proteins that interact with desmosomal proteins have also been implicated in ARVC pathogenesis [1,12]. Mutations in these genes disrupt desmosomal integrity, making muscle fibers susceptible to tearing, fragmentation and eventually cell death in the course of the cardiac cycle. Loss of desmosomal function also affects gap junction remodeling, sodium channel function, and electrocardiographic parameters in cardiomyocytes [1]. In addition, perturbation of desmosomal proteins promotes adipogenesis in mesodermal precursors by suppressing the Wnt/β-catenin signaling pathway, which serves important functions in cardiac myogenesis [37,38,39]. These result in fibro-fatty replacement of the ventricular myocardium, predominantly in the RV.

### 2.4. Left Ventricular Non-Compaction Cardiomyopathy

LVNC is a heterogeneous disorder characterized by prominent trabeculae, a thin compacted layer, and deep intertrabecular recesses most evident in the LV apex [40]. Non-compaction may involve the RV, presenting as either a biventricular or isolated RV non-compaction phenotype. The genetic form of LVNC is commonly inherited as an X-linked recessive or autosomal dominant condition [1,40]. Autosomal recessive and mitochondrial inheritance for LVNC have also been reported. Inherited LVNC is attributed to mutations affecting compaction of the endomyocardial layer that normally progresses from the base to the apex of the heart during embryogenesis [12]. Mutations in multiple genes have been identified to cause or contribute to the development of LVNC. These include genes encoding for sarcomeric (e.g., *MYH7*), Z-disc (e.g., *LDB3*), nuclear envelope (e.g., *LMNA*), mitochondrial (e.g., *TAZ*), and ion channel proteins (e.g., *SCN5A*) [12]. 

### 2.5. Restrictive Cardiomyopathy

RCM is the least common form of cardiomyopathy, typically manifesting as having increased ventricular stiffness that impairs ventricular filling in the absence of ventricular hypertrophy or systolic dysfunction [2,41]. Most disease-causing mutations are inherited in an autosomal dominant fashion, although autosomal recessive, X-linked and mitochondrial inheritance forms also exist [12]. Alterations in genes encoding for sarcomeric proteins (e.g., *TNNT2*), Z-disc proteins (e.g., *MYPN*) or transthyretin (*TTR*) have been identified in patients with RCM [42]. 

### 2.6. Cardiomyopathy in DMD and Other Disorders

The cardiomyopathy in DMD manifests as DCM and develops in three stages [43]. In the first stage, occurring before the teens, DMD patients show no symptoms of heart failure. The heart presents with some hypertrophy, resulting in diastolic dysfunction. Electrocardiogram (ECG) abnormalities are apparent [44]. This progresses to the clinical stage, where the heart progressively dilates and accumulates fibrosis. Fibrosis typically begins at the inferobasal wall of the LV, which spreads with age [44]. Over 90% of DMD patients will have manifested cardiac dysfunction after 18 years of age [44,45]. Finally, the last stage is characterized by end-stage heart failure: the DCM in DMD patients has reached its most severe state, having systolic dysfunction. Arrhythmias are common. Interestingly, female carriers of *DMD* mutations may also manifest DCM—one study found that 8% of female DMD carriers had this phenotype [46]. This has the potential to lead to severe heart failure in some cases, and so must be closely monitored.

Cardiomyopathy is likewise observed in diseases such as Marfan and Barth syndromes. Marfan syndrome is an autosomal dominant disorder with an estimated prevalence of 1:5000 births [47]. It is caused by mutations in the *FBN1* gene that codes for fibrillin-1. Fibrillin-1 is an extracellular protein that helps form microfibrils, which, among other functions, provide elastic properties to tissues [48]. DCM is typically associated with Marfan syndrome. It is debated whether the dilated phenotype is caused by volume overload related to valvular insufficiency (mitral valve prolapse is commonly seen in patients) or by defects in the cardiac muscle itself [47,49]. On the other hand, Barth syndrome is an X-linked autosomal recessive disorder affecting around 1:300,000–400,000 births, and is caused by mutations in the tafazzin (*TAZ*) gene [50]. Tafazzin is important for the synthesis of cardiolipin, a phospholipid found in both inner and outer mitochondrial membranes that functions in mitochondrial protein transport, cellular respiration, and mitophagy regulation besides its structural role [50,51]. Loss of tafazzin and hence cardiolipin results in compromised energy stores, decreased contractility and increased damage for the heart [52]. Barth syndrome cardiomyopathy is usually DCM, but cases of HCM and potential LVNC have been described [51,53]. 

## 3. Genome Editing for Cardiomyopathies

### 3.1. Creating Disease Models 

Unlike other disciplines, reliable disease models are relatively lacking in the cardiovascular field [13]. In vitro models, such as cardiomyocyte cell lines, are not readily available, while in vivo models such as rodents do not faithfully recapitulate the presentation of cardiovascular diseases in humans. Other models, such as rabbits or nonhuman primates, better resemble human disease course, however, they are difficult to maintain and the application of genome editing has not yet been that successful or attempted in these models. Non-mammalian models of cardiomyopathies are also available, such as in zebrafish, *C. elegans*, and *Drosophila*. Despite having more physiological and anatomical dissimilarities with the human heart, they have proven useful for understanding cardiac development, regeneration, and the pathophysiology of certain cardiovascular disorders [54,55,56]. These models, particularly *Drosophila*, have also been adapted for testing variants of unknown significance (VUSs), leading to patient diagnosis [57,58,59]. Moreover, the ease of acquiring large sample sizes with these models allows for high-throughput screening of candidate therapies [55]. Genome editing has not yet been used for model creation in these systems, however, and a careful understanding of their limitations in the study of cardiomyopathies is needed for the interpretation of obtained results.

Previous studies have looked into generating human induced pluripotent stem cell (hiPSC) lines from patients owing to their ability to be differentiated into various cell types. This is particularly beneficial for therapies targeting the heart as patient-derived hiPSCs can easily be induced to differentiate into cardiomyocytes. These induced cardiomyocytes (iCMs) can then be used to explore the cardiac-specific effects of treatment on specific patient genetic backgrounds [60,61]. Advances in genome editing have revolutionized the cardiovascular field, allowing the creation of isogenic cell lines differing only at the locus of interest. From a therapeutic standpoint, rodents and dogs also exhibit increased tolerance to drugs than humans, and thus have limited utility in assessing cardiotoxicity. In contrast, hiPSCs exhibit 70%–90% accuracy in detecting cardiotoxicity [62]. Multiple gene-editing tools have been used to create disease models for cardiomyopathies and a selected few studies will be discussed in the following sections.

#### 3.1.1. In Vitro Models

TALENs have been used to generate a wide variety of human embryonic stem cell (hESC)-based models to study cardiomyopathies. Karakikes and colleagues (2017) used TALEN constructs to knockout 88 genes associated with cardiomyopathies and congenital heart diseases [63]. All constructs were validated and found to disrupt the target locus at high frequencies. TALEN pairs targeting the start codon or DNA regions immediately downstream are more efficient compared to TALEN pairs targeting the 5′-UTR. To determine the utility of these models for studying molecular mechanisms underlying cardiomyopathies, the authors examined TALEN mutants of the cardiac troponin T (*TNNT2*) gene. *TNNT2* mutations are commonly implicated in autosomal dominant HCM and sometimes DCM. The authors generated both heterozygous knockout (TNNT2+/−) and homozygous knockout (TNNT2−/−) iPSC lines. The TNNT2 homozygous knockout model generated by TALENs showed hallmark features of cardiomyopathy, such as sarcomeric disarray and impaired intracellular Ca^2+^ cycling, while the heterozygous knockout did not show any structural or functional abnormalities suggesting that haploinsufficiency is unlikely to explain the pathogenesis of *TNNT2*-related cardiomyopathies. Correcting a dominant-negative mutation in the *TNNT2* gene also ameliorated the DCM phenotype, validating the utility of these models. 

Various cardiomyopathy models have been created using CRISPR [64,65,66]. Using patient-derived hiPSCs, CRISPR/Cas9, and tissue engineering, the pathophysiology of Barth syndrome cardiomyopathy was replicated in tissue constructs [64]. Phenotypic rescue by gene replacement and small molecule treatments has also been demonstrated using this model. In another study by Mosqueira et al. (2018), CRISPR/Cas9 was used to create 11 isogenic variants of an HCM-causing mutation in the *MYH* gene in three independent hiPSC/hESC lines, which were subsequently differentiated into cardiomyocytes for molecular and functional assessment [65]. These cardiomyocytes showed the main features of HCM at the cellular level such as hypertrophy, excessive multi-nucleation, and sarcomeric disarray. Functional characterization demonstrated energy depletion, Ca^2+^ handling abnormalities, arrhythmias, and hypo-contractility. The pharmacological rescue of arrhythmias was shown to be feasible. Additionally, novel long non-coding RNAs (lncRNAs) and putative gene modifiers were identified using these models, which unravels new therapeutic avenues for HCM.

#### 3.1.2. In Vivo Models

Genome-editing technologies have been used to generate animal models to study cardiomyopathies [67,68,69,70,71]. One of the most promising approaches entails the use of somatic in vivo genome editing, in which genome editing tools are delivered into a target organ in an adult animal. Carroll et al. (2016) described the generation of cardiomyocyte-specific Cas9 transgenic mice expressing high levels of Cas9 in the heart without other systemic abnormalities [67]. Intraperitoneal injection of adeno-associated virus serotype 9 (AAV9) encoding gRNA against *Myh6* in postnatal cardiac-Cas9 transgenic mice resulted in effective genome editing in cardiac tissue. These mice displayed cardiomyopathy and heart failure as demonstrated by aberrant cellular changes consistent with HCM and elevated heart failure markers, respectively. While this method overcomes embryonic lethality and allows tissue-specific genome editing, a previous study indicated that the effect of postnatal cardiac genome editing using this approach is target-dependent [69]. Further studies are necessary to expand the versatility of CRISPR/Cas9 somatic genome editing as a tool to study cardiomyopathy and other cardiovascular diseases in general.

Advances in genome editing technology enable the generation of genetically-modified animals other than mice [68,70]. Using ZFNs and somatic cell nuclear transfer (SCNT) technologies, Umeyama and colleagues (2016) generated a heterozygous fibrillin-1 (*FBN-1*) mutant pig model of Marfan syndrome [68]. The model showed clinical features consistent with Marfan syndrome such as skeletal and aortic abnormalities. A pig model of HCM was also created using SCNT and TALENs, in this case to introduce a mutation into the *MYH7* gene [70]. Although the mutant pigs exhibited HCM features such as cardiomyocyte disarray, malformed nuclei and *MYH7* overexpression, the animals died within 24 h postpartum. These studies highlight both the potential and challenges associated with genome editing in large animal models for the study of cardiomyopathy.

### 3.2. Studying Disease Pathophysiology

Genome editing has been used to help elucidate the underlying pathophysiological mechanisms of cardiomyopathies. For DCM, targeted mutations in the gene of the sarcomeric protein titin revealed that truncation of its C-terminal region causes severe myopathy whereas mutations in its N-terminal region exhibit milder phenotypes [72]. A conserved internal promoter that partially rescues the N-terminal truncations was also identified, explaining the divergence in disease severity between N- and C-terminal mutations. A seminal study by Hinson et al. (2015) demonstrated that sarcomeric insufficiency is the underlying cause of DCM [73]. Using iCMs with different mutations in the *TTN* gene, the study revealed pathogenic missense variants that diminish contractile performance. Truncations in the I-band domain of titin were better tolerated than truncations in the A-band domain due to alternative exon splicing. Mutant titin protein in iCMs impairs sarcomeric structure and myofibrillar assembly. The resulting sarcomeric insufficiency causes impaired cardiomyocyte responses to mechanical and adrenergic stress. Reduced growth factors and cell signaling activation are also observed, both of which are essential processes in adaptive remodeling of the myocardium. iCMs have also been used to investigate the role of *BAG3* in DCM, by studying two mutants: R477H known to cause DCM and a *BAG3* knockout. In this study, McDermott-Roe et al. (2019) found that both mutants showed myofibrillar disarray and impaired protein quality control in CMs [74]. 

Similarly, genome editing has been applied to study the role of various genes and proteins in the pathogenesis of HCM. Using iCMs and CRISPR/Cas9, Jaffre and colleagues (2019) were able to recapitulate the HCM phenotype and identify the molecular mechanisms by which mutations in the *RAF1* gene cause HCM in individuals with Noonan syndrome (NS) [75]. They examined iCMs derived from the reprogrammed fibroblasts of an NS patient with a *RAF1^S257L^*^/+^ mutation. Activation of mitogen-activated protein kinase kinase 1/2 (also known as MEK1/2), but not extracellular-signal-regulated kinase 1/2 (ERK1/2), induced the formation of abnormal cardiomyocyte structure whereas extracellular-signal-regulated kinase 5 (ERK5) caused an enlarged cardiomyocyte phenotype. In another study, CRISPR/Cas9 and TALENs were used by Seeger et al. (2019) to investigate the molecular mechanisms underlying HCM associated with mutations in the *MYBPC3* gene that introduce premature stop codons [76]. iCMs containing these mutations exhibited aberrant Ca^2+^ handling and other molecular dysregulations not due to haploinsufficiency of the MYBPC3 protein. The nonsense-mediated decay pathway was discovered to play an essential role in HCM pathogenesis. Genome editing has also been used for assessing variants of unknown significance (VUSs) in the context of genetic screening for HCM [77]. 

Genome editing has likewise contributed to our understanding of the less common cardiomyopathies such as LVNC and ARVC [78,79]. Kodo et al. (2016) recapitulated the LVNC phenotype in iCMs carrying a mutation in the gene for the cardiac transcription factor TBX20 [78]. The proliferative defects associated with LVNC were identified to be a consequence of abnormal TGF-β signaling activation. In addition, CRISPR/Cas9 technology was used to characterize mutations in the *SCN5A* gene coding for Nav1.5 sodium channels in the context of ARVC [79]. The study found reduced sodium currents as well as decreased Nav1.5 and N-cadherin clusters at junctional sites in the mutant model, suggesting Nav1.5 is in a functional complex with cell adhesion molecules. These results provide an alternative explanation to the mechanisms by which *SCN5A* mutations cause ARVC. Genome editing has also been used to further our understanding of essential processes in the pathophysiology of cardiomyopathy such as cardiomyocyte maturation and cardiac remodeling, among others [80,81].

### 3.3. Therapeutic Genome Editing

Despite advances in cardiovascular as well as pharmaceutical research, treatment for cardiomyopathies remains limited. Genome editing has the potential to modulate the expression of the gene of interest, offering a novel avenue for therapeutic treatment. This approach was applied to the phospholamban gene (*PLN*), the protein product of which functions to regulate the kinetics of calcium flux in cardiomyocytes [82,83,84]. Mutations in *PLN* have been implicated in the development of cardiomyopathies. iCMs harboring a deleterious *PLN* R14del mutation exhibited irregular Ca^2+^ handling, abnormal cytoplasmic distribution of phospholamban protein and increased expression of cardiac hypertrophy markers [82]. The R14del mutation was corrected using a TALEN vector pair designed to introduce a double-strand break adjacent to the mutation and the gene correction matrix designed to incorporate the wild-type copy of the gene into the DNA via recombination. Genetic correction of the mutation resulted in functional phenotypic correction as shown by normalized calcium handling, regression of the hypertrophic phenotype and homogeneous reticular distribution of phospholamban. In a follow-up study using three-dimensional human engineered cardiac tissue technology, the *PLN* R14del mutation was found to impair cardiac contractility; TALEN-mediated genetic correction restored contractile function in this model [84].

Recent developments in precise genome-editing techniques have enabled the correction of germline mutations in humans. In a recent study that sparked much discussion in the scientific community and the general public at large, Ma et al. (2017) described the correction of a heterozygous *MYBPC3* mutation associated with HCM in human pre-implantation embryos using CRISPR/Cas9-mediated homology-directed repair (HDR) pathway with an endogenous, germline-specific DNA repair response. [85]. CRISPR/Cas9 introduces double-stranded DNA breaks which are preferentially resolved by the error-prone non-homologous end-joining (NHEJ) pathway. HDR pathway is an alternative option that repairs the double-strand break using the wild-type copy of the gene or a supplied exogenous DNA molecule as template, leading to correction of the mutant allele. However, the efficiency of HDR is relatively low. Remarkably, the study showed that DSBs in the mutant paternal allele were predominantly repaired through HDR. HDR was exclusively directed by the maternal non-mutant homologous copy, suggesting human embryos employ different DNA repair mechanisms compared to somatic or pluripotent cells. A major problem with genome editing in human embryos is mosaicism, which was investigated and overcome in the study by co-injecting sperm and CRISPR/Cas9 into metaphase II oocytes. Even though further studies are needed before clinical applications, these results clearly demonstrate that CRISPR/Cas9 has potential to be used for the correction of heritable mutations in human embryos. As the technology is now beginning to make its entry into human studies, the ethical considerations surrounding genome editing have to be discussed and reviewed as well.

## 4. Genome Editing for Cardiomyopathy in Duchenne Muscular Dystrophy

Genome editing strategies to understand or treat cardiomyopathies part of more systemic disorders are also being studied. A number of groups have already generated in vitro and in vivo systems to more closely model and comprehend cardiac involvement in disorders such as Marfan syndrome, Barth syndrome, and Fabry disease [64,66,68]. In terms of treatment, genome editing has arguably seen the most progress in its development as a therapy for DMD. The majority of therapeutic approaches being developed for DMD typically fail to address its cardiac aspects. Antisense therapy, while certainly at the forefront, has historically been impeded by its lack of efficacy in the heart [6]. The field is just beginning to overcome this through the introduction of more efficient cardiac delivery methods [86], but it will take time before these are optimized for both safety and efficacy. Moreover, current management strategies are only capable of slowing down cardiac failure, not prevention [87]. It is highly encouraging that novel genome editing strategies are being developed for DMD that have the potential to treat its associated cardiomyopathy, aside from addressing defects in skeletal muscle.

Here, we discuss studies using CRISPR technology to treat DMD, as it is the most widely used genome editing approach in the field. The main goal is to restore the reading frame of the mutated *DMD* gene using CRISPR, allowing for the synthesis of partially functional, truncated dystrophin protein [88]. The approaches typically fall into four categories. Three of which rely on NHEJ repair, and result in either reframing of a frame-disrupting exon, the deletion of a single out-of-frame exon, or the deletion of multiple out-of-frame exons. The fourth one relies on HDR, and uses an exogenous template to correct specific mutations. We focus on those that present cardiac findings, with implications for the treatment of DMD-associated cardiomyopathy. For a more general discussion of CRISPR studies on DMD, we direct the reader to our previous review [88]. 

### 4.1. Studies Using Human iPSC Models

Studies using hiPSCs for developing CRISPR DMD treatments are summarized in Table 1. The use of hiPSCs is recently emerging for DMD therapy research and carries with it various advantages. For instance, CRISPR can be used to generate desired patient mutations in control hiPSC lines, providing unprecedented versatility in modeling a great number of *DMD* mutations in vitro. iCMs have the added advantage of more capably modeling the human heart in terms of physiology [62,89]. Differences in ion channels responsible for repolarization, as well as in the localization of sarcomeric proteins exist between mouse and human hearts, to name a few. A striking example would be how mild the cardiac phenotype is in *mdx* mice [90], one of the most widely used animal models for DMD. iCMs from DMD patient hiPSCs are also able to model particular disease phenotypes quite well, including having significantly greater cell areas, longer resting sarcomere lengths and a decreased capacity to respond to environmental stimuli than control iCMs [91]. Impaired calcium handling and contraction dynamics are likewise observed in DMD iCMs [62]. Overall, these features lend hiPSCs well to therapeutic development, allowing one to determine if treatments can lead to potential improvements in cardiac function. This is exactly what studies developing CRISPR therapies for DMD capitalized on, with five out of seven evaluating cardiac phenotypes post-treatment (Table 1).

One of the earliest studies using DMD iCMs for CRISPR treatment was that by Young and colleagues in 2016 [92]. Young et al. used hiPSCs derived from the fibroblasts of patients with deletions in either *DMD* exons 46–51 or 46–47, or with duplication of *DMD* exon 50. The strategy involved using CRISPR/SpCas9 to delete exons 45–55 from the *DMD* gene, which should restore the reading frame in all patient hiPSCs used. Exons 45–55 are located in one of two mutation hot spots in the *DMD* gene, the exons 43–55 distal hot spot, where 73% and 76% of all *DMD* deletions in patients begin and end, respectively [93]. It is estimated that skipping exons 45–55 will help treat approximately 66% of all DMD patients with deletion mutations [93].

After nucleofecting plasmids containing SpCas9 and gRNAs against *DMD* introns 44 and 55 into hiPSCs, the treated hiPSCs were differentiated to become iCMs. These iCMs showed the expected skipping of exons 45–55, as well as dystrophin rescue via Western blotting and immunocytochemistry (ICC). To test membrane integrity, treated and non-treated DMD iCMs (from Δex46–51 and Δex46–47 patient hiPSCs) were exposed to hypo-osmotic conditions, and the amount of CK released into the medium was measured. Treated iCMs showed reduced CK release levels than dystrophic iCMs, at a level similar to healthy controls, demonstrating the functionality of the produced dystrophin in stabilizing cell membranes.

Kyrychenko et al. (2017) also looked into the feasibility of deleting multiple exons to treat DMD [95]. The authors investigated three approaches, either deleting *DMD* exons 3–9, exons 6–9, or exons 7–11, all of which are in-frame deletions, using CRISPR/SpCas9. These are all part of the proximal *DMD* gene mutation hot spot, extending from exons 1 to 22. Compared to exons 43–55, a lower but still considerable number of mutations occur in this region, with 23% and 16% of all *DMD* deletions beginning and ending here, respectively [93]. Using an hiPSC line with an engineered *DMD* exons 8–9 deletion, they found that deleting exons 3–9 not only restored dystrophin production but also offered the most improvement in terms of Ca^2+^ kinetics and synchronicity in Ca^2+^ activity in iCMs. Engineered heart muscle (EHM) from exons 3–9-deleted iCMs displayed the most enhanced contractility as well. Similar results were obtained when exons 3–9 were deleted in patient-derived hiPSCs with an exons 3–7 deletion. Exons 3–9 skipping is estimated to treat ~7% of DMD patients with out-of-frame deletions [93]. A study by Nakamura et al. (2016) reported that of the combined 15 patients that have a *DMD* exon 3–9 deletion in the Leiden Open Variation Database, the universal mutation database UMD-DMD, and in two case studies, 11 were either BMD patients or asymptomatic [100]. This suggests that the deletion of exons 3–9 likely results in the production of stable, partially functional dystrophin, further highlighting this approach as a favorable strategy for therapeutic genome editing.

The rest of the studies were mainly concerned with the removal or skipping of single *DMD* exons. Diverse genome editing approaches were used, which required changes in gRNA targets and/or the Cas enzyme used. Long and colleagues (2018) focused on correcting three different patient mutations using a variety of strategies that use regular Cas9 [96]. Besides single exon skipping, they designed approaches to eliminate a pseudoexon through removal of a cryptic splice site as well as to remove a duplicate segment of exons using a single gRNA. All strategies rescued dystrophin synthesis, with corresponding improvements in the force of contraction of EHMs generated from treated iCMs. By mixing together corrected and uncorrected iCMs and using these to make EHM models, the authors found that at least 30% of cardiomyocytes need to be corrected to partially rescue the dystrophic phenotype in vitro. At least 50% correction was required for complete rescue, as expected from the X-linked recessive nature of DMD.

Yuan et al. (2018) used a catalytically deficient version of Cas9 (dCas9) fused to a cytidine deaminase to specifically edit single DNA bases [97]. Applying this to eliminate the exon 50 donor splice site in exon 51-deleted patient hiPSCs, they achieved exon 50 skipping and dystrophin production in iCMs. Improved performance in the hypo-osmotic stress test was observed in treated iCMs. Finally, Zhang et al. (2017) used a different CRISPR enzyme, Cpf1 instead of Cas9, to reframe/skip exon 51 in patient hiPSCs with an exons 48–50 deletion [94]. Cpf1 works similarly to Cas9, with some differences in gRNA structure, PAM site preference, and producing a sticky rather a blunt end after DNA cleavage [26]. The approach was able to rescue dystrophin production and significantly increase mitochondria copy numbers and oxygen consumption rates compared to control non-treated iCMs.

Indeed, iCMs from hiPSCs are useful models for testing the functional efficacy of DMD CRISPR therapies in the context of a relevant cell type. However, hiPSCs do have three main limitations. First, a given hiPSC population is subject to multiple sources of variation [62]. There is genomic variation from differences in reprogramming, phenotypic variation from differences in how hiPSCs are differentiated into iCMs, cellular variation from how hiPSCs can differentiate into more than one cardiomyocyte subtype (e.g., nodal, ventricular, and atrial), and inter-laboratory variation from differences in protocols. Methods for standardizing the reprogramming and differentiation of hiPSCs are currently being optimized to address these issues. Second, iCMs actually exhibit limited maturation [62,89,101]. In fact, it has been shown that iCMs are more similar to fetal than adult cardiomyocytes, and thus may not faithfully recapitulate all disease phenotypes. Third, while they offer a more physiologically similar study system to humans, they lack the physiological context which animal models are able to provide. Advances in producing EHM using 3D cardiac scaffolds are underway, but methods that allow for more diverse functional assessments of therapeutic effects using these models are lacking and are still being developed [89]. Studies on the pharmacological behavior of administered genome editing agents are also more appropriately done in an in vivo system.

The utility of hiPSCs in testing emerging genome editing therapies cannot be denied. However, results should be interpreted in conversation with in vivo data to provide a more comprehensive view of therapeutic performance. In response to the limitations that exist when using either hiPSCs or animal models, more groups are now using both systems in developing their own genome editing therapies. With this complementary approach, the strengths of one can cover for the weaknesses of the other.

### 4.2. Studies Using Animal Models

In 2014, Long et al. aimed to correct the mutant *Dmd* gene in *mdx* mice, a mouse model of DMD [102]. These mice have a spontaneous nonsense point mutation in *Dmd* exon 23, which leads to a lack of dystrophin [103]. The approach entailed injecting 1-cell *mdx* embryos with Cas9 mRNA, a gRNA for exon 23, and a single-stranded oligodeoxynucleotide template for HDR. Eleven progeny with corrected *Dmd* genes were obtained, seven corrected by HDR and four corrected by NHEJ repair. The percentage of cells carrying corrected genes varied across mice. At 7–9 weeks of age, around 40%–80% dystrophin-positive fibers were observed in the heart of these corrected mice, depending on the level and type of correction present. Lowest dystrophin levels were found with 17% HDR, and the highest levels were found with 83% NHEJ. While this study certainly demonstrated the promise of CRISPR genome editing as a therapy for DMD, it did not provide much information as to whether the approach would be useful for treating the dystrophic heart. No assessments of cardiac function were performed, and we do not know what minimum levels of correction are needed to achieve ameliorative effects on the cardiac phenotype. Germline editing is also currently not a feasible option of treatment, considering the many technical and ethical issues associated with it.

In the next five years, much in vivo work would be done in developing CRISPR as a viable therapy for DMD. Addressing the need to treat not only the skeletal muscles but also the heart in DMD is becoming increasingly recognized as well. Table 2 summarizes a list of in vivo DMD CRISPR studies that have looked into the effects of genome editing treatment on the heart. Essentially three kinds of animal models were used: the *mdx* mouse and its variants, CRISPR-generated dystrophic mice, and the deltaE50-MD dog. Interestingly, one of the *mdx* variants used (del52hDMD/*mdx*) carries a stably integrated mutant human DMD transgene, which allows for the in vivo testing of gRNAs targeting human sequences [104].

All but one of the studies in Table 2 were done using mouse models, with the majority on correcting the genetic defect in *mdx* mice [94,102,105,106,107,111,112,113]. It is difficult to make comparisons of therapeutic efficacy across studies, not only because of all the variations in experimental design and genome editing strategy but also because not all of these studies provided quantitative results. In addition to this, there are inter-study differences in the method of dystrophin rescue quantification to account for—stressing the need to develop globally standardized methods. Despite these limitations, some general impressions can be made.

Dystrophin rescue in the heart was observed after CRISPR treatment in all studies, either through Western blotting or immunohistochemistry (IHC). Rescue ranged from <1% to 94% of healthy dystrophin levels in Western blot, and about the same percentage range of dystrophin-positive fibers in IHC. The level of dystrophin rescue appears to be independent of the route/age of administration and the viral vector used. An exception would be the case when adenoviruses are used for delivery, since they do not seem to be capable of penetrating beyond the periphery of the heart [109]. Dystrophin restoration, at least in the heart, does not appear to be strongly influenced by whether a dual or single vector system is used for CRISPR component delivery.

On the other hand, dose has a direct influence on efficacy. Higher doses of transduced Cas9 and gRNA vectors usually lead to increased dystrophin rescue in the heart, as well as in skeletal muscles. In one study, tripling the dose of a single Cas9-gRNA vector increased the number of dystrophin-positive fibers to about 40%, when initially there were only a few, scattered dystrophin-positive fibers present in the heart [109]. Min et al. (2019) published an interesting study on dosing, which showed that the ratio of Cas9 to gRNA vector amounts administered in vivo was a critical determinant of therapeutic efficacy [99]. In the study, they treated mice carrying a deletion in *Dmd* exon 44 through the CRISPR-mediated skipping or reframing of exon 45. Mice at post-natal day 4 (P4) were injected with AAV9 vectors carrying SpCas9 at a constant dose of 5 × 1013 vg/kg and AAV9 vectors carrying gRNA at varying ratios, from 1 to 10 times the Cas9 dose. The percentage of dystrophin-positive fibers in the heart strikingly rose from 10% at 1:1 Cas9:gRNA vector to 94% at 1:10 Cas9:gRNA vector, on average.

This corroborated results from an earlier study by Hakim et al. (2018), which found that the gRNA vector genome was preferentially depleted in vivo compared to the Cas9 vector genome [111]. In that study, the authors similarly demonstrated that increasing the amount of gRNA vector provided to *mdx* mice, at a 1:3 Cas9:gRNA vector ratio, led to increased dystrophin rescue in the heart post-treatment. It is surmised that the limiting property of gRNAs is related to how the gRNA vector adopts an unstable hairpin/cruciform structure in solution, how gRNAs are critical in helping stabilize the conformation of Cas9 to its active form, how fast gRNA turnover is, or how increasing gRNA vector dose ensures that more nuclei express gRNA and are thus amenable to being acted upon by Cas9 [99,111]. Whichever the reason, further investigating this relationship may lead to substantial improvements in the development of CRISPR therapies.

Most in vivo studies developing CRISPR treatments for DMD are done within short timelines. Evaluations on therapeutic efficacy are typically performed 3 to 14 weeks post-injection, which may not provide a sufficient length of time to appreciate the long-term benefits of treatment. Considering that these studies also begin injecting mice at relatively young ages (P1 to 11 weeks-old), and with most using the *mdx* model, it is no wonder that the capacity of CRISPR therapy in improving cardiac function could not be evaluated. One study says that *mdx* mice start showing signs of mechanical cardiac dysfunction at around 18 months of age [90], highlighting the need for more longitudinal assessment. This is unless, of course, the *mdx* model used carries a genetic background that predisposes it to earlier cardiac symptoms as in the case of *mdx*/*Utr*^+/-^ mice. These mice have deficient production of utrophin, a dystrophin homolog thought to compensate for dystrophin loss in mdx mice [114]. El Refaey et al. (2017) used this model for CRISPR therapy testing, and successfully saw improvements in the contractility of isolated papillary muscles as early as 10 weeks post-treatment [109].

Recently, in response to the above limitations, three studies were published evaluating the long-term therapeutic efficacy of DMD CRISPR treatment in mdx mice [111,112,113]. All aimed to delete exon 23 from the *Dmd* gene, either by itself or along with exons 21 and 22, producing an in-frame transcript for dystrophin translation. Assessment of therapeutic success was performed from 12 to 19 months post-injection. The restoration of dystrophin in the heart was persistent across the three, with up to 20% dystrophin of healthy levels observed at 18 months post-treatment in one study by Western blotting [111]. Dystrophin-positive fibers were observed in all studies by IHC. Importantly, two studies showed that CRISPR treatment led to significant improvements in cardiac function, based on electrocardiography and echocardiography [111,112]. These show the promise of CRISPR for treating DMD-related cardiomyopathy in patients.

CRISPR therapies have also been tested in a dog model of DMD. In a study by Amoasii et al. (2018), two dogs carrying out-of-frame deletions in *Dystrophin* exon 50 were each injected with different doses of Cas9- and gRNA-carrying AAV9 vectors [98]. One dog was given a low 2 × 10^13^ vg/kg/vector dose, and the other was given a higher 1 × 10^14^ vg/kg/vector dose. The strategy was to reframe or skip exon 51 in these dogs by a single-cut NHEJ approach targeting the exon 51 splice acceptor site. Eight weeks post-injection, 92% dystrophin of healthy levels were observed in the heart of the dog injected with the high dose by Western blotting, as well as dystrophin-positive fibers by IHC. Cardiac dystrophin rescue levels were not quantified for the dog that received the lower dose, however noticeably fewer dystrophin-positive fibers were found overall in the heart compared to the high-dose dog. It would be interesting to see evaluations of cardiac function in the future, as DMD dog models typically phenocopy patient symptoms better than mouse models [115]. In line with this, a related thrust in the field is to use CRISPR to develop other, more phenotypically representative DMD animal models. Rat [116], rabbit [117], and pig [118] models have already been developed, and all of these present with cardiac features reminiscent of DMD. These can be used for testing future DMD therapies, not just for genome editing, but also for those aiming to make improvements in the dystrophic heart.

## 5. Recent Advances in CRISPR Genome Editing with Potential for Cardiomyopathy Research

Efforts seeking to improve the state of genome editing for investigating or treating cardiomyopathies and other genetic disorders are actively ongoing. Next-generation sequencing, bioinformatics analyses, clinical discoveries, and basic research are constantly identifying new genes involved in cardiomyopathies, providing novel therapeutic targets for genome editing [119,120]. This includes genes that give rise to non-coding RNA, e.g., over 1000 lncRNAs have been found to be dysregulated in HCM [121], and various microRNAs involved in cardiac remodeling and regeneration [122,123]. As we have learned, new in vitro and in vivo models are also being generated at a steady pace, a process expedited through genome editing. Aside from offering insights into the biology of cardiomyopathies, these models can serve as platforms for testing genome editing therapies in unique contexts.

We focus primarily on advances in CRISPR, as this is the currently favored technique and most research is going into the improvement of this technology. An area of ongoing work is on how to enhance HDR rates for CRISPR in the heart. HDR only occurs in the S and G2 phases of the cell cycle [124]. As the heart essentially consists of post-mitotic cells, the predominant mode of DNA repair is NHEJ, which may be unfavorable if the intended strategy is gene replacement or knock-in. Additionally, the unpredictable nature of NHEJ in making indels is a safety concern due to its potential in introducing off-target effects. A number of strategies have been devised to suppress NHEJ and enhance HDR repair. These include the use of small molecules, e.g., Src7 for DNA ligase IV inhibition [125,126], and a variety of genetic/molecular strategies, e.g., shRNA knockdown of KU70 and KU80 [126], genome editing in combination with cell cycle synchronization [127], and the use of geminin-Cas9 fusion proteins [128].

Although effective in vitro, work is needed to translate these approaches in vivo as certain genetic manipulations are simply not possible in a larger, animal system [13,129]. Continued research on pathways involved in DNA repair such as the Fanconi anemia pathway [130] may prove useful in developing other strategies for enhancing HDR. Considering there is evidence for HDR occurring in cardiomyocytes in vitro despite being non-actively dividing cells [131], developing a method to shift the balance from NHEJ to HDR in the heart is highly feasible.

Another relatively recent advancement is a new genome editing strategy, homology-independent targeted integration (HITI), developed by Suzuki and colleagues in 2016 [132]. HITI operates like a hybrid of NHEJ and HDR, as it is capable of gene knock-in but does so using NHEJ. The main advantage offered by HITI is its efficiency in both actively proliferating and non-dividing cells. HITI used with CRISPR/Cas9 was capable of integrating a green fluorescent protein (GFP) construct into mouse primary neurons in vitro, with 55.9% of transfected cells being GFP-positive. In contrast, an HDR-based technique yielded almost no GFP knock-ins. The same situation was found in vivo, with HITI-CRISPR/Cas9 able to knock-in GFP into the brain and skeletal muscles. The method was also applied to correct the *Mertk* gene in rats with retinitis pigmentosa, with treated retinas showing significantly improved phenotype and function. Interestingly, intravenously administered HITI-CRISPR/Cas9 has been shown to be capable of knocking-in genes in the heart, at a rate significantly higher than HDR. It would be most intriguing to see if HITI can be applied for treating cardiomyopathies in the future.

Cas enzyme efficacy, specificity, and versatility are constantly undergoing improvement. Classical SpCas9 enzymes are being engineered to enhance such properties, and we now have a considerable selection to choose from for genome editing, e.g., eSpCas9(1.1) [133], SpCas9-HF1 [134], HypaCas9 [135], evoCas9 [136], and xCas9 [137], among others. These variants are usually created by substituting specific amino acids in the original Cas9 protein to improve specific binding affinity to target sequences. This could also help expand the number and kind of PAM sites recognized by the engineered Cas enzyme, as in the case of xCas9. A high-fidelity version of SaCas9 called SaCas9-HF has been created recently as well [138]. There is also a variant of the Cas9 enzyme that is catalytically-inactive (dCas9), which can be fused to transcriptional activators or repressors, allowing for the control of gene expression at an “epigenetic” level [139]. Base editors, which we have discussed, make use of dCas9 as well. A growing interest in the field is the search for other Cas enzymes. Not only would this expand the list of possible PAM sites (and hence list of genes) that can be recognized, the discovery of novel enzymes could also introduce new strategies for genome editing. Furthermore, this is important when it comes to packaging the Cas gene into viral vectors, as this has been an issue in the therapeutic delivery of SpCas9. Cas9 from other bacterial species have been described [140,141], as well as enzymes from other CRISPR/Cas types and classes [142].

Strides have also been made in trying to reduce off-target effects associated with genome editing, particularly for CRISPR. Strategies can be broadly divided into two, depending on which CRISPR component is modified. One set of approaches focuses on the Cas enzyme. Aside from engineering Cas9 itself, groups have also divided the enzyme in half so that a double-strand DNA break can only be accomplished when both halves are at the target site [143,144]. These so-called paired nickases have the potential to be more efficient than the original, single enzyme [145]. Self-restricting mechanisms to reduce Cas9 transcription and/or translation have also been devised, through the co-administration of gRNAs against the genome of the delivery vector or the use of synthetic repression systems [146,147,148]. The other set of approaches aims to enhance gRNA design. Optimizations of gRNA sequence, length, and chemistry have all been performed, with the development of bioinformatics tools further helping with gRNA screening [149,150,151,152,153]. The creation of more robust and comprehensive methods to evaluate off-target effects in the genome such as GUIDE-seq [154] and Digenome-seq [155] further facilitates efforts in alleviating the concern associated with the safety of genome editing for therapy.

Finally, continued advances in delivery are making the heart more amenable to genome editing. New viral vectors are being made, e.g., a protease-activatable AAV based on AAV9 has recently been demonstrated to specifically deliver transgenes to the heart in a mouse model of myocardial infarction [156]. The current trend, however, is in the development of non-viral delivery methods for genome editing. Non-viral delivery overcomes issues of immunogenicity, unwanted viral genome integration, and limited packaging seen with viral vectors. Lipid nanoparticles, polymer-based particles, cell-penetrating peptides, DNA nanoclews, and inorganic nanoparticles (silicon- or zinc-based), among others, are examples of non-viral approaches that have been used for delivering genome editing agents [157,158,159]. An exciting development is the use of gold nanoparticles for CRISPR/Cas9 delivery (CRISPR-Gold), which has been applied for correcting the *Dmd* point mutation in *mdx* mice via HDR [160]. Whether CRISPR-Gold exhibits effective targeting to the heart remains to be seen. Application of these non-viral delivery approaches to the heart may help improve research into cardiomyopathies.

## 6. Conclusions

The global burden of cardiomyopathy to health is unarguably high. From the records of one center spanning roughly a 30–35 year period, more than 50% of sudden cardiac deaths and cardiac transplantations were attributed to cardiomyopathy [161]. This stresses the need for conducting research on this group of cardiovascular disorders, a need genome editing is increasingly addressing. Despite their immense heterogeneity, many groups have successfully identified genes linked to cardiomyopathies—information that can be exploited for or provided by genome editing. This availability of genetic information holds true for cardiomyopathies part of more systemic myopathies, a prime example being DMD. After a brief survey of the field, we have now seen how genome editing has advanced not only our knowledge on the various cardiomyopathies, but also the development of therapies for these disorders.

On that note, with genome editing having already been conducted on human embryos to correct a cardiomyopathy-related gene mutation [85], the field is certainly heading into using this technology for therapeutic purposes. As promising as it is, genome editing still has a number of challenges to overcome in this regard, which we can summarize as issues of efficacy or safety. Efficacy is primarily tied to strategy design, delivery, and endpoint assessment. Design is constantly improving, owing to the development of bioinformatics tools and genome editing enzymes, among others. Studies on delivery have been limited, however, owing to the more prevalent use of in vitro hiPSC systems for research. Interestingly, studies from DMD models suggest that the heart is surprisingly well-favored for viral vector-based delivery of genome editing agents [99,105,106,107,110,111,112,113]. It would be interesting to see if this is the case for other in vivo cardiomyopathy models or how non-viral means of delivery may compare. Methods for evaluating the therapeutic efficacy of genome editing techniques would definitely require standardization, e.g., in the case of DMD, the method used largely determines what percentage of dystrophin rescue is quantified (Table 2). Since the development of genome editing treatments for primary cardiomyopathies is still in its early stages this should be kept in mind.

Safety mostly relates to the specificity of the genome editing approach, as well as toxicity resulting from the therapy itself. On-target (viral genome integration) and off-target (editing of non-target sequences) mutagenesis are the biggest concerns. However, advances in delivery as well as strategy design, respectively, are helping mitigate these issues. The same advances are helping reduce toxicity resulting from unwanted immune responses, for instance by using engineered delivery vectors “invisible” to the immune system or through perhaps designing gRNAs that will be less susceptible to activating innate immune responses [157,158,162,163,164].

Over the past decade, genome editing has rapidly been being incorporated into the biomedical field. The use of CRISPR/Cas9 offers insight into fundamental biological processes as well as the potential of curing and preventing heritable human diseases. However, these advances raise considerable ethical and social issues [165,166]. Genome editing in germline cells or embryos has been seen as controversial, mostly because we still do not fully understand its consequences (such as off-target effects, immune response activation) in humans [167,168]. Concerns regarding the clinical application of genome editing also require further consideration, e.g., who are the candidates for genome editing, when would it be applied, which diseases merit treatment through this approach, and its legal implications. The future of genome editing will necessitate active discussion between scientists, clinicians, government institutions, and the general public, among others, to consider all potential ethical repercussions.

As with any therapy, genome editing requires a strict balance between efficacy and safety, maximizing therapeutic benefit while minimizing patient risk. With further development, accompanied by the constant rise in our knowledge on these disorders, it is only a matter of time until this balance is achieved for the treatment of inherited cardiomyopathies.

## Figures and Tables

**Table 1 ijms-21-00733-t001:** Clustered regularly interspaced short palindromic repeats (CRISPR) studies on Duchenne muscular dystrophy (DMD) treatment using induced cardiomyocytes (iCMs) from human induced pluripotent stem cell (hiPSCs).

Model	Strategy, Nuclease	Delivery, Vector/s	Cardiac Findings				Reference
			*RT-PCR*	*WB*	*ICC*	*Function, etc.*	
Δex46–51, Δex46–47, ex50dup. from patient fibroblasts	*DMD* Δex45–55, SpCas9	Nucleofection, dual plasmid	Skipping observed	DYS observed	DYS^+^ cells observed	iCMs from treated iPSCs (Δex46–51, Δex46–47) had reduced CK release in hypo-osmotic conditions	2016 Young et al. [92]
Δex48–50 from patient fibroblasts	*DMD* ex51 NHEJ repair or skipping, LbCpf1/AsCpf1	Nucleofection, single plasmid	Reframing, skipping observed	DYS observed in all strategies	DYS^+^ cells observed in all strategies	iCMs from reframed iPSCs had significantly more mitochondria and increased respiratory capacity	2017 Zhang et al. [94]
Δex8–9, CRISPR-generated from healthy PBMCs in study; Δex3–7 from patient (source not stated)	*DMD* Δex3-9, Δex6-9, Δex7–11, SpCas9	Nucleofection, dual plasmid	Skipping observed in all strategies	DYS observed in all strategies (Δex7–11 had least DYS)	DYS^+^ cells observed in all strategies	Ca^2+^ dynamics improved after treatment, but only significant in Δex3–9 iCMs; EHM from treated iCMs had enhanced contractility, with Δex3–9 showing best results	2017 Kyrychenko et al. [95]
Δex48–50, pseudo-ex47, ex55–59dup. from patient PBMCs	*DMD* ex51 skipping, cryptic splice site removal in ex47, Δ55–59dup., respectively, SpCas9	Nucleofection, single plasmid	Skipping observed in all strategies	DYS observed in all strategies	DYS^+^ cells observed in all strategies	EHMs from corrected iCMs had significantly improved contractile force; 30% or 50% DYS^+^ CMs sufficient for partial or complete recovery, respectively	2018 Long et al. [96]
Δex51 from patient PBMCs	*DMD* ex50 skipping, dSaCas9-TAM	Lipotransfection, single plasmid (with separate Ugi plasmid)	Skipping observed	DYS observed	DYS^+^ cells observed	iCMs from treated iPSCs had significantly reduced CK release in hypo-osmotic conditions	2018 Yuan et al. [97]
Δex48–50 from patient fibroblasts	*DMD* ex51 NHEJ repair or skipping, SpCas9	Nucleofection, single plasmid	-	67%–100% DYS of WT observed	DYS^+^ cells observed	-	2018 Amoasii et al. [98]
Δex44 from patient PBMCs	*DMD* ex43, 45 NHEJ repair or skipping, SpCas9	Nucleofection, single plasmid	-	DYS observed	DYS^+^ cells observed	-	2019 Min et al. [99]

Abbreviations: RT-PCR, reverse transcription-polymerase chain reaction; WB, Western blot; ICC, immunocytochemistry; DYS, dystrophin; CK, creatine kinase; NHEJ, non-homologous end-joining; PBMCs, peripheral blood mononuclear cells; EHM, engineered heart muscle.

**Table 2 ijms-21-00733-t002:** CRISPR studies on DMD treatment using animal models, with cardiac-related findings.

Model	Strategy, Nuclease	Delivery, Vector/s (Viral Dose If Available)	Observation Period	Cardiac Findings				Reference
				*RT-PCR*	*WB*	*IF*	*Function, etc.*	
*mdx*	*Dmd* ex23 HDR/NHEJ repair, SpCas9	1-cell embryo injection, Cas9 mRNA/gRNA/ssODN	7–9 weeks	-	DYS observed	~40%–80% DYS^+^ fibers	-	2014 Long et al. [102]
*mdx*	*Dmd* Δex23, SpCas9	RO at P18, dual AAV9 (1.8 × 10^13^ vg *)	4, 8, 12 weeks post-injection	Skipping observed	DYS observed (8, 12 wk.)	1.1%–9.6% DYS^+^ fibers (71.1% of WT)	-	2016 Long et al. [105]
	*Dmd* Δex23, SpCas9	IP at P1, dual AAV9 (6.0 × 10^12^ to 1.0 × 10^13^ vg *)	4, 8 weeks post-injection	-	-	1.1%–3.2% DYS^+^ fibers (52.4% of WT)	-	
*mdx*	*Dmd* Δex23, SaCas9	IP at P2, dual AAV8 (2.8 × 10^11^ vg/vector)	7 weeks post-injection	Skipping observed (more than TA, ~DIA)	-	Few DYS^+^ fibers	-	2016 Nelson et al. [106]
	*Dmd* Δex23, SaCas9	IV at 6-wks, dual AAV8 (2.7 × 10^12^ vg/vector)	8 weeks post-injection	Skipping observed	>6.25% DYS of WT observed	Many scattered DYS^+^ fibers	-	
*mdx;Ai9*	*Dmd* Δex23, SaCas9	IP at P3, dual AAV9 (1.5 × 10^12^ vg/vector)	3 weeks post-injection	~5% skipping observed	<1% DYS of WT observed	Few DYS^+^ fibers	-	2016 Tabebordbar et al. [107]
	*Dmd* Δex23, SaCas9	IV at 6-wk., dual AAV9 (3.6 × 10^13^ vg/vector)	14 weeks post-injection	>10% skipping observed	<1% DYS of WT observed	Few DYS^+^ fibers	-	
*mdx^4cv^*	*Dmd* Δex52–53, SpCas9/SaCas9	RO at 11-wks, dual AAV6 (low dose, 1 × 10^12^ vg/vector; high dose, 1 × 10^13^ vg Cas9, 4 × 10^12^ vg gRNA) or single AAV6 (1 × 10^12^ vg)	4 weeks post-injection	-	DYS observed, more at high dose	Up to 34% DYS^+^ fibers, widespread	-	2017 Bengtsson et al. [108]
*mdx*	*Dmd* ex51 HDR repair, LbCpf1	1-cell embryo injection, Cpf1 mRNA/gRNA/ssODN	4 weeks	-	DYS observed	DYS^+^ fibers increasing with HDR correction	-	2017 Zhang et al. [94]
*mdx*/*Utr^+^*^/*-*^	*Dmd* Δex21-23, SpCas9	IV/IP at P1-3, single AdV (~2.5 × 10^10^ vg)	4 weeks post-injection	Skipping observed	DYS observed	DYS^+^ fibers only at peripheral myocardium	-	2017 El Refaey et al. [109]
	*Dmd* Δex21-23, SaCas9	RO/IP at P3, single AAVrh74 (low dose, 3 × 10^11^ vg; high dose, 1 × 10^12^ vg)	10 weeks post-injection	Skipping observed	23.3% DYS of WT observed at high dose	DYS^+^ fibers observed, ~40% at high dose	Contractility significantly improved post-treatment; β-adrenergic responsiveness not affected	
	*Dmd* Δex21-23, SaCas9	IV at 16-wks, single AAVrh74 (1 × 10^12^ vg)	7 days post-injection	-	-	DYS^+^ fibers observed	-	
ΔEx50 mice, CRISPR-generated in study	*Dmd* ex51 NHEJ repair or skipping, SpCas9	IP at P4, dual AAV9 (6.3 × 10^10^ vg *)	4, 8 weeks post-injection	Reframing, skipping observed	DYS observed	Widespread DYS^+^ fibers	-	2017 Amoasii et al. [110]
del52h*DMD*/*mdx*	Hybridization of *DMD* ex47 and 58, SaCas9	IV at 4/5-wks, dual AAV9 (3.75 × 10^13^ vg/kg/vector)	6 weeks post-injection	Hybridization observed (not in TA or DIA)	DYS observed	DYS^+^ fibers observed	-	2018 Duchêne et al. [104]
deltaE50-MD dog	*Dystrophin* ex51 NHEJ repair or skipping, SpCas9	IV at 1-mo, dual AAV9 (low dose, 2 × 10^13^ vg/kg/vector; high dose, 1 × 10^14^ vg/kg/vector)	8 weeks post-injection	Skipping observed	92% DYS of WT observed at high dose	DYS^+^ fibers observed, increasing with dose	-	2018 Amoasii et al. [98]
*mdx*	*Dmd* Δex23, SaCas9	IV at 6-wks, dual AAV9 (1^st^ study, 7.2 × 10^12^ vg Cas9, 3.63 × 10^12^ vg gRNA; 2^nd^ study, 1 × 10^13^ vg Cas9, 3 × 10^13^ vg gRNA)	8, 18 months post-injection	Skipping observed in both studies	5% DYS of WT at 18 mo., study 1; 20%/9% DYS of WT in males/females at 18 mo., study 2	DYS^+^ fibers observed in both studies	Study 1: ECG showed significant improvement at 18 mo.; Study 2: ESV, EF, ECG, hemodynamics improved in treated females at 18 mo., no functional data for males	2018 Hakim et al. [111]
ΔEx44 mice, CRISPR-generated in study	*Dmd* ex45 NHEJ repair or skipping, SpCas9	IP at P4, dual AAV9 (5 × 10^13^ vg/kg Cas9, various for gRNA)	4 weeks post-injection	-	94% DYS of WT at 1:10 Cas9:gRNA dose	94% DYS^+^ fibers at 1:10 Cas9:gRNA dose	-	2019 Min et al. [99]
*mdx*	*Dmd* Δex21-23, SaCas9	IP at P3, single AAVrh74 (1 × 10^12^ vg)	19 months post-injection	-	2.16% DYS of WT observed	11.1% DYS^+^ fibers observed	CO and SV (echo) significantly improved post-treatment, with reduced levels of cardiac troponin I	2019 Xu et al. [112]
*mdx*	*Dmd* Δex23, SaCas9	IV at P2, dual AAV8/9 (5.4 × 10^11^ vg/vector)	1 year post-injection	>50% skipping observed	DYS observed	DYS^+^ fibers observed	-	2019 Nelson et al. [113]

* unsure if vg/vector or total vg dose; Abbreviations: RT-PCR, reverse transcription-polymerase chain reaction; WB, Western blot; IF, immunofluorescence; HDR, homology-directed repair; NHEJ, non-homologous end-joining; ssODN, single-stranded oligodeoxyribonucleotide; DYS, dystrophin; RO, retro-orbital; IP, intraperitoneal; IV, intravenous; WT, wild-type; AAV, adeno-associated virus; TA, tibialis anterior; DIA, diaphragm; ECG, electrocardiogram; ESV, end-systolic volume; EF, ejection fraction; CO, cardiac output; SV, stroke volume

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
