# Peer review of "Genome Editing for the Understanding and Treatment of Inherited Cardiomyopathies"

_ijms, 2020, doi:10.3390/ijms21030733_

Round 1

Reviewer 1 Report

The review manuscript entitled "Genome editing for the understanding and treatmentof cardiomyopathies" by Nguyen, Lim and Yokota focuses on recent technological developments in genome editing applied to modeling and treating diverse cardiomyopathies. My opinion is that this manuscript provides a valuable cardiomyopathy-dedicated compedium of knowledge on how quickly evolving genome editing field helps in creating cardiac disease models and in studying disease pathophysiology. Review is well-written, comprehensively organized and disusses all major aspects and all major published data related to in vitro (iPSCs) and in vivo mammalian models. One suggestion I have is to mention in the chapter "Creating disease models" that other non-mammalian simple and genetically amenable animal models of cardiomyopathies also exist including zebrafish and Drosophila (for review Duncker et al., 2015 and Taghli-Lamallem et al., 2016). Importantly, genome editing is well developed in these  simple models and applied for modeling and testing undiagosed variants (VUS) identified by genome sequencing of patients with different rare diseases including those with cardiac defects. I think, Undiagnosed Disease Network (UDN) that is in charge of this important initiative could be cited in the review (several papers - Splinter et al., 2018, Bellen et al., 2019).

Minor comment:

there is a duplicated sentence line 186/187 and 189/190

Author Response

Reviewer 1

One suggestion I have is to mention in the chapter "Creating disease models" that other non-mammalian simple and genetically amenable animal models of cardiomyopathies also exist including zebrafish and Drosophila (for review Duncker et al., 2015 and Taghli-Lamallem et al., 2016). Importantly, genome editing is well developed in these  simple models and applied for modeling and testing undiagosed variants (VUS) identified by genome sequencing of patients with different rare diseases including those with cardiac defects. I think, Undiagnosed Disease Network (UDN) that is in charge of this important initiative could be cited in the review (several papers - Splinter et al., 2018, Bellen et al., 2019).

Response: Thank you for your comment. We have added the following statements to the section of concern: “Non-mammalian models of cardiomyopathies are also available, such as in zebrafish, C. elegans, and Drosophila. Despite having more physiological and anatomical dissimilarities with the human heart, they have proven useful for understanding cardiac development, regeneration, and the pathophysiology of certain cardiovascular disorders [62–64]. These models, particularly Drosophila, have also been adapted for testing variants of unknown significance (VUSs), leading to patient diagnosis [65–67]. Moreover, the ease of acquiring large sample sizes with these models allows for high-throughput screening of candidate therapies [63]. Genome editing has not yet been used for model creation in these systems, however, and a careful understanding of their limitations in the study of cardiomyopathies is needed for the interpretation of obtained results.”

There is a duplicated sentence line 186/187 and 189/190

Response: Thank you for pointing this out. We have removed the duplicated sentence as requested.

Reviewer 2 Report

The review "Genome editing for the understanding and treatment of cardiomyopathies" by Nguyen and coleagues adresses a currently hot topic and might therefore be of interest to the scientific community. The manuscript is nicely written and gives a good overwiev on genome editing in cardiomyopathies. Still, I have a few questions and comments that should be addressed:

1.) As the authors correctly state, cardiomyopathies are very heterogenous. In the here provided classification the big field of the ischemic cardiomyopathies (ICM) is missing. Although some people count ICM into the DCM group, the pathomechanisms are completely different and therefore shouold be addressed separately. As the ICM is not of genetic origin it is probably not too important for this manuscript, but should at least be mentioned in the introduction and/or discussion.

2.) If the authors do not address ICM on the manuscript, the title should probably phrased a bit more precisely.

3.) The abbreviation for the disease (DMD) is the same as for the dystrophine gene (DMD). Although the gene being italizised, it is a bit confusing while reading the paper. Is there any  possibility to change this?

4.) On page 5, lines 185-187 the scentence "This is particular beneficial for therapies targeting the hart as patient derived hiPSCs can easily be induced..." is repeated a few lines later. This should be corrected.

5.) Page 15, lines 539-546. What was the outcome of the dog that got the lower vector dose? This should be stated, othewise the reference does not make sense.

6.) Genome editing is a still very controversly discussed technique. There are regional und cultural differences, especially when it comes to humans. One paragraph about ethical concernes should be added to the manuscript.

Author Response

Reviewer 2

As the authors correctly state, cardiomyopathies are very heterogenous. In the here provided classification the big field of the ischemic cardiomyopathies (ICM) is missing. Although some people count ICM into the DCM group, the pathomechanisms are completely different and therefore shouold be addressed separately. As the ICM is not of genetic origin it is probably not too important for this manuscript, but should at least be mentioned in the introduction and/or discussion.

Response: Thank you for the comment. We agree with the reviewer regarding the non-genetic origin of ICM, and this is the reason why we chose not to talk about it in the manuscript. Thus, we have modified the title as suggested (see response to comment 2 below) to better clarify our subject matter.

If the authors do not address ICM on the manuscript, the title should probably phrased a bit more precisely.

Response: Thank you for the suggestion. Since we have decided not to address ICM, we have instead changed the title to “Genome editing for the understanding and treatment of inherited cardiomyopathies”

The abbreviation for the disease (DMD) is the same as for the dystrophine gene (DMD). Although the gene being italizised, it is a bit confusing while reading the paper. Is there any  possibility to change this?

Response: Thank you for this comment. The use of DMD as an abbreviation for the disease and DMD as an abbreviation for the dystrophin gene is a widely-used convention in the field and is common practice in most papers. As such, we do not think it possible to change this situation.

On page 5, lines 185-187 the scentence "This is particular beneficial for therapies targeting the hart as patient derived hiPSCs can easily be induced..." is repeated a few lines later. This should be corrected.

Response: Thank you for pointing this out. We have removed the duplicated sentence as requested.

Page 15, lines 539-546. What was the outcome of the dog that got the lower vector dose? This should be stated, otherwise the reference does not make sense.

Response: Thank you for this comment. The study did not report any cardiac muscle-related improvements for the dog that got the lower vector dose. To inform readers of this, we have added the following statement in the paragraph: “Cardiac dystrophin rescue levels were not quantified for the dog that received the lower dose, however noticeably fewer dystrophin-positive fibers were found overall in the heart compared to the high-dose dog.”

Genome editing is a still very controversly discussed technique. There are regional und cultural differences, especially when it comes to humans. One paragraph about ethical concerns should be added to the manuscript.

Response: Thank you for this thoughtful comment. We have added the following paragraph to the conclusion: “Over the past decade, genome editing has rapidly been being incorporated into the biomedical field. The use of CRISPR/Cas9 offers insight into fundamental biological processes as well as the potential of curing and preventing heritable human diseases. However, these advances raise considerable ethical and social issues [174,175]. Genome editing in germline cells or embryos has been seen as controversial, mostly because we still do not fully understand its consequences (such as off-target effects, immune response activation) in humans [176,177]. Concerns regarding the clinical application of genome editing also require further consideration, e.g. who are the candidates for genome editing, when would it be applied, which diseases merit treatment through this approach, and its legal implications. The future of genome editing will necessitate active discussion between scientists, clinicians, government institutions and the general public, among others, to consider all potential ethical repercussions.”